# Time-Driven Activity-Based Costing for Capturing the Complexity of Healthcare Processes: The Case of Deep Vein Thrombosis and Leg Ulcers

**DOI:** 10.3390/ijerph20105817

**Published:** 2023-05-13

**Authors:** Carla Rognoni, Alessandro Furnari, Marzia Lugli, Oscar Maleti, Alessandro Greco, Rosanna Tarricone

**Affiliations:** 1Centre for Research on Health and Social Care Management (CERGAS), SDA Bocconi School of Management, Bocconi University, 20136 Milan, Italy; alessandro.furnari@unibocconi.it (A.F.); rosanna.tarricone@unibocconi.it (R.T.); 2National Reference Training Center in Phlebology (NRTCP), Vascular Surgery—Cardiovascular Department, Hesperia Hospital, 41125 Modena, Italy; lugli@chirurgiavascolaremodena.com (M.L.); maleti@chirurgiavascolaremodena.com (O.M.); 3Outpatient Wound Care Centre, Local Health Care System, 03100 Frosinone, Italy; a.greco2009@gmail.com; 4Department of Social and Political Science, Bocconi University, 20136 Milan, Italy

**Keywords:** venous stenting, standard medical treatment, compression therapy, deep venous outflow obstruction, leg ulcers, cost-utility analysis, time driven activity-based costing

## Abstract

Time-driven activity-based costing (TDABC) is suggested to assess costs within the value-based healthcare approach, but there is a paucity of applications in chronic diseases such as deep vein thrombosis (DVT) and leg ulcers. In this context, we applied TDABC in a cost-effectiveness analysis comparing venous stenting to compression ± anticoagulation (standard of care—SOC) from both hospital and societal perspectives in Italy. TDABC was applied to both treatments to assess costs that were included in a cost-effectiveness model. Clinical inputs were retrieved from the literature and integrated with real-world data. The Incremental Cost Utility Ratio (ICUR) of stenting compared to SOC was EUR 10,270/QALY and EUR 8962/QALY for hospital and societal perspectives, respectively. The mean cost per patient for venous stenting of EUR 5082 was higher than the Diagnosis-Related Group (DRG) reimbursement (EUR 4742). For SOC, an ulcer healing in 3 months costs EUR 1892, of which EUR 302 (16%) is borne by the patient versus a reimbursement of EUR 1132. TDABC showed that venous stenting may be cost-effective compared with SOC but that reimbursement rates may not completely cover the real costs, which are partially sustained by the patients. A more efficient policy for covering the real costs may be beneficial for both clinical centers and patients.

## 1. Introduction

Globally, hospitals, governments, and payers are under tremendous pressure to improve patient clinical outcomes while reducing costs. Thus, “value-based healthcare” was developed as a healthcare delivery framework to address the challenges facing healthcare [1]. Value is defined as “health outcomes achieved per unit cost expended over the entire care delivery value chain” [2], ignoring boundaries between departments and organizations and capturing all processes in the care continuum for a medical condition. In this context, time-driven activity-based costing (TDABC) represents the best way to capture the complexity of healthcare processes and translate them into costs [2,3,4]. Furthermore, it calculates the total cost of staff and clinical resources involved at each step implemented in clinical centers and, consequently, the cost of the patient’s entire care cycle.

The literature reports various applications of the TDABC methodology, but there is a paucity of studies in the context of chronic diseases such as the management of patients with deep vein thrombosis and specifically leg ulcers.

Venous leg ulcers are due to sustained venous hypertension resulting from chronic venous insufficiency. Venous pressure remains high in a system where the valves are inefficient. In chronic venous insufficiency and post-thrombotic syndrome (PTS) after deep vein thrombosis (DVT), valve damage causes venous reflux from the deep to the superficial circulation, with venous hypertension and stasis. Approximately 10% of the population in Europe and North America has valve insufficiency, and 0.2% of this population develop venous ulcers [5]. Additionally, 45% of venous ulcers are due only to superficial venous insufficiency and/or perforating ones with the presence of a normal deep venous system. Many risk factors such as DVT, varicose veins, and obesity cause the onset of venous ulcers, and 70% of those at risk develop them [5].

Compression combined with anticoagulant therapy is the standard of care (SOC) for venous ulcer management [6]. The pressure exerted must be graduated; approximately 30–40 mmHg at the ankle, which reduces to 15–20 mmHg at the calf, is generally adequate for healing most venous leg ulcers. Different compression systems may be used. These include multilayer elastic compression bandages, long or short-stretch bandages, or elastic tubular bandages. Compression is also available with pneumatic devices [7]. In Italy, these products are not reimbursed by the NHS so they are bought by the patients themselves [8].

Endovenous stenting has emerged as a new option to treat iliofemoral venous outflow obstruction. An endovenous stent may be defined as a synthetic tubular structure implanted in native or graft vasculature to provide mechanical radial support and enhance vessel patency. Actually, more than ten dedicated venous stents are available on the European market; the main differences are related to the design, the material, the deployment system, and the different sizes [9,10,11,12,13,14,15]. The design can be closed or open cells with different shape and size of the cell area. The material used in new generation venous stents is nitinol or even elgiloy (material composed of cobalt, chromium, nitinol, iron, etc.). Another technical feature is the different modality of deployment in the vein to guarantee the precision and stability of the stent: it can be performed in a pull-back technique or using a rotating thumbwheel; some systems are even re-constrainable, allowing for the change and adjustment of the landing zone for the stent. A wide range of sizes is actually available (10–24 mm diameter, 40–160 mm length).

Stenting is used in patients with established PTS after the previous DVT to reduce symptoms of chronic pain and swelling and to aid ulcer healing in severe cases. Venous stenting may also be used to improve symptoms of obstruction in patients presenting with acute DVT to prevent the development of PTS.

In Italy, endovascular stenting for DVT and leg ulcers is reimbursed by the NHS through DRG (Diagnosis-Related Group) 479 with a tariff at national level of EUR 4742 [16].

The significant advances in stent design for venous circulation led to a low risk of morbidity and mortality. Nevertheless, data are lacking on long-term outcomes [11]. Although clinical guidelines provide recommendations in order to manage DVT disease best, their implementation varies across countries from strict adherence to no adherence at all [17], further contributing to ineffective management of DVT cases. 

To our knowledge, only the ongoing C-TRACT randomized controlled trial is comparing endovascular stenting versus SOC [18]. However, real-world data from retrospective or prospective observational studies and registries are available and may provide an assessment of both the care and health outcomes for patients in routine clinical practice, especially in medical devices [19,20,21]. 

Real-world evidence has also been employed for evaluating the cost-effectiveness of venous stenting compared to SOC; the literature currently reports a study in the US [22] and Italy [23]. The former considered real-world data on 9 patients with stenting and 17 patients with compression and evaluated the real costs from the hospital perspective and quality of life data for the two treatment strategies. The study showed that iliac stenting for venous leg ulcers was less costly (USD 1913 per patient) and increased the quality of life (0.01 [Quality Adjusted Life Years] [QALYs]) compared with compression alone over a time horizon of 3 years. The second study performed a meta-analysis on the rates of healed and recurring ulcers and showed that stenting is a cost-effective (incremental cost-utility ratio [ICUR] EUR 12,388/QALY) or dominant option versus SOC from the national healthcare service perspective for in-patient or day-hospital settings for stenting, respectively.

The purpose of this study was to provide a cost-effectiveness analysis for the comparison of venous stenting with SOC in Italy using the TDABC methodology, considering both hospital and societal perspectives. The precise estimation of healthcare resources used and related costs along the care pathway process may provide further evidences to support stakeholders in evaluating the available therapy options related to managing patients with DVT and leg ulcers. 

## 2. Materials and Methods

This analysis started from the Italian hospital perspective with identifying the production and cost functions for the provision of healthcare services for the treatment of leg ulcers (stenting procedure or SOC) and patient follow-up in the clinical practice. The process steps associated with both treatments for the clinical management of leg ulcers were identified and mapped through field observations and interviews. 

Once the patient clinical pathways were defined and validated by clinicians, the TDABC methodology was used to assess the cost of each process along the care continuum delivery path [3]. The possible costs related to the disease sustained by the patients along the process of ulcer management (e.g., for the purchase of compression devices) were also factored in to provide a comprehensive view of the total medical costs. For stenting procedures, we referred to an Italian hospital, Hesperia Hospital (Modena), while for the SOC management of ulcers and patient follow-up, we interviewed a clinical expert.

For the cost-effectiveness analysis, we used the model structure already presented by Rognoni and colleagues [23] which compared stenting procedures versus SOC. Clinical outcomes (rates of leg ulcers healed and recurred) were updated according to a literature review.

All methods were carried out in accordance with relevant guidelines and regulations (PRISMA for systematic reviews and CHEERS for the economic evaluation), and the data collection protocol was approved by the involved clinical center (Hesperia Hospital).

### 2.1. Literature Review

The authors of the present study conducted in July 2019 a literature review [23] that reported the rates of healed and recurring ulcers for venous stenting and SOC. Given that the reported rates may be outdated, we updated the literature review from July 2019 to April 2021 using the same search strategy. The PRISMA diagrams on our literature review are presented in Appendix B. 

### 2.2. Clinical Data Synthesis

A total of 187 and 322 studies were identified for SOC and venous stenting, respectively. After duplicates removal and screening of studies, four and two studies were included for quantitative synthesis for SOC and stenting procedure, respectively. The studies on SOC considered Unna boot and compression systems.

Appendix A show the study characteristics of the included studies for SOC and stenting, respectively. Appendix A includes also the updated real-world data on stenting regarding the 88 patients affected by PTS and ulcers, which was collected at Hesperia Hospital in Modena, Italy (study Lugli-Longhi-Maleti). These real-world data reported 88 active ulcers with an 81% healing rate (mean healing time 2 months) and 20 recurrences (23% on the initial number of ulcers) considering a follow-up of 41 months.

The review update confirmed the lack of head-to-head trials comparing the two treatment options, and consequently, a meta-analysis was performed for stenting and SOC using single-arm data on the percentage of ulcer healing and recurrence. Recurring ulcers were reported as a percentage of the number of active ulcers. For SOC, the percentage of ulcers healed was 61% (mean healing time: 3.3 months), with a 10% recurrence rate; these values for stenting were 81% (mean healing time: 3 months) and 5%, respectively. Higher heterogeneity among studies was reported for ulcers healed with SOC. Overall, SOC and stenting studies reported a maximum follow-up of 24 and 41 months, respectively. 

Meta-analyses were performed using Stata software (metaprop command) using a random-effect model [24]. Appendix A report the updated forest plots obtained from all the studies from the observation period. 

### 2.3. TDABC

For stenting, process mapping involves hospitalization in a regular admission setting and activities characterizing the surgical procedure. Briefly, the TDABC steps involved (1) developing process maps for care delivery pathways; (2) measuring the time required for each process step and determining capacity cost rates for staff and clinical resources activated; and (3) calculating the total cost of care delivery. 

Thus, the TDABC had two sequential steps:Process mapping and recording time data: three rounds of data gathering were performed. Firstly, we conducted an online interview due to mobility restrictions during the COVID-19 pandemic to identify the main phases and activities that characterize surgical procedures. The interview was conducted on 3 June 2021 and included a vascular surgeon and her staff. In particular, the vascular surgeon interviewed was the head of the Hesperia surgical team, with over 20 years of experience. Her staff consisted of a second vascular surgeon, the nursing coordinator, and two administrative assistants. During the interview, we asked them to describe all activities involving staff and patient flows from patient admission to its discharge. For each activity, we therefore asked to indicate the execution time, the kind of personnel involved, goods and services consumed, spaces and technologies used. A draft of a process map was developed, which specified resources (personnel and clinical) required in each step of the care cycle. Additionally, it included an estimation of the time of each resource used. Secondly, we conducted a face-to-face interview with the vascular surgeon. This step was critical to validate the process map and time estimation and to gather possible missing data. This second interview was conducted on 12 July 2021 at Hesperia Hospital. Thirdly, we conducted an in-depth observation of the processes mapped by following a patient during her hospitalization, from admission to discharge. This allowed us to compare data gathered through interviews with the actual care activities, solving possible misalignment in activities, sequence, and timing. This third step was implemented between 10 and 11 November 2021 at Hesperia Hospital.Costs data gathering and utilization: Once the processes and resources were mapped, the cost data collection phase was initiated. We defined a template and the personnel of the administrative office at Hesperia Hospital supported us in identifying and isolating the cost of personnel, goods and services, spaces and technology. The capacity cost rate for each resource and process step was calculated as the euro per minute capacity cost rate for all the clinical resources absorbed by the process steps. For healthcare personnel, costs were computed by multiplying the minutes spent in each phase by the wage per minute for each role (details about the capacity costs for the different roles and timings applied are reported in Appendix A). Finally, the cost of caring for the patient in the total care cycle was estimated.

Thus, all cost information over the total cycle of care, including personnel, equipment, and facilities, as well as the capacity cost of each resource used during patient’s care and the cost associated with staff functions, such as information technology and administration, were tracked. 

Similarly, we replicated the TDABC steps above and estimated the total costs for SOC. First, we identified the care phases in which patients were involved and the resources absorbed by each phase. Specifically, we included the initial visit for managing leg ulcers, follow-up visits, and the final management phase in ulcer healing. Resources and timing data were collected by interviewing an Italian clinical expert (Dr. Alessandro Greco). Four online semi-structured interviews were conducted between November and December 2021. After validation of each phase, we gathered cost data for the pathway. Costs analyses (December 2021, EUR) considered costs sustained by the hospital and the patients (see Appendix A for details on healthcare personnel costs).

### 2.4. Cost-Effectiveness Model

The analysis was reported according to Consolidated Health Economic Evaluation Reporting Standards (CHEERS) [25,26]. The CHEERS checklist is reported in Appendix C.

The cost information was included in an already developed cost-effectiveness model [18] that compared stenting procedure versus SOC. The model relied on clinical outcomes like rates of leg ulcers healed and recurred. In this setting, the systematic literature search of these outcomes [23] was updated and integrated with an extended set of real-world data on venous stenting. The Markov model, already developed by Rognoni and colleagues [23], was updated with the new clinical parameters on the rates of healed and recurred ulcers reported in the meta-analyses. The model was constructed with an adult population of patients with DVT and leg ulcers with health states “active ulcer”, “healed ulcer”, and “recurred ulcer”. Costs were updated using the TDABC costs for stenting and SOC to represent the hospital and social perspectives. The time horizon remained 36 months, in line with the study duration used in the meta-analyses. A discount rate of 3% was applied to the QALYs and costs [27] and the 1-month Markov cycle length was maintained. The healing times of 2 and 3 months for stenting and SOC, respectively, were considered. The model details are reported in [23]. Appendix A shows the patients’ distributions (Markov cohort analysis) over time among the health states for the SOC and stenting.

The model allowed to estimate mean costs and QALYs for the management of a patient with the two options, stenting procedure or SOC, over a time horizon of 36 months.

### 2.5. Cost-Effectiveness Analysis

Cost-effectiveness analysis was expressed through the incremental cost-utility ratio (ICUR), calculated as the difference in the mean expected costs divided by the difference in the mean expected QALYs between stenting procedure and SOC, as obtained by the model:ICUR = (Cost _Stenting_ − Cost _SOC_)/(QALYs _Stenting_ − QALYs _SOC_)

One-way and probabilistic sensitivity analyses (PSA) were performed to test the robustness of the model by varying the parameters by ±20% of their baseline values. The PSA was conducted through 1000 Monte Carlo simulations by assigning distributions to model parameters (beta for utilities/percentages, and gamma for costs/healing times, with a standard deviation of 20% of the baseline value). Results were presented graphically as an acceptability curve. Appendix A reports the details of the parameters used in these analyses.

In order to visually represent the outcome and cost data simultaneously, we referred to a radar chart [28,29]. This graph provides a snapshot of the value being delivered by a specific medical condition and allows a comparison of the values across the treatment options: stenting and SOC. The radar chart includes the costs obtained through TDABC and axis labels that are of interest to stakeholders: cost borne by the hospital, cost borne by the patient, QALYs, rate of ulcer healing, rate of no recurrence, and rate of procedural success. Outcome data points for each treatment modality were graphed on separate axes, where all axes ranged from 0 to 100. Cost data were reported as normalized relative cost ratios anchored to the lowest cost in the study. Normalized ratios were incorporated into the diagram as reciprocals to allow for consistency in interpreting the radar chart where improved outcomes or lower costs were indicated by data points farther from the center of the graph [29].

## 3. Results

Using the TDABC approach to endovascular procedure, we identified 9 macro-phases and 36 micro-phases in the stenting care cycle. We identified three preliminary macro-phases (patient acceptance, preliminary assessment, and pre-operative hospitalization), three surgical macro-phases (preparation for surgery, surgical intervention strictly considered, and surgery closure), and three post-operative macro-phases (post-operative hospitalization, post-operative assessment, and discharge) (Figure 1).

Table 1 shows the total time and costs per macro-phase. The blocks of preliminary (44%) and post-operative (45%) macro-phases absorb more time than other phases. Conversely, the block of operative macro-phase absorbs more costs (83% of the total costs) compared to other macro-phases. Thus, considering a stenting procedure in a regular hospital admission, the mean cost per patient was EUR 5082. We observed that stent cost represented the most important component of non-personnel costs, with an average of EUR 1420 and a mean number of stents used per procedure of 1.5. The stent cost represents the 55% of the total goods, services, tech, and infrastructural costs. Another relevant cost item for the stenting procedure at Hesperia Hospital is the use of advanced technology such as IVUS, which accounts for 18% of the total goods, services, tech, and infrastructural costs. A “loan for use agreement” as a contractual form for machine acquisition was in place, which we considered were for probes and drug consumption. During data gathering, we observed variability factors such as patients’ weight or complexity in terms of clinical conditions and included them proportionally to the percentage of patients characterized by the single factor. Thus, the cost values reported in Table 1 considered the intrinsic variability of the reference cases.

The analysis also accounted for variations in procedural success (97% from meta-analysis, see Appendix A). Thus, we explored extra costs for complications, which caused a re-intervention in 3% of the cases, leading to a total cost of EUR 5234 for the stenting strategy.

Similarly, we estimated the costs for SOC after identifying the main phases of the care cycle. Table 2 highlights the phases and costs for patient care management, distinguishing the first visit, subsequent monthly visits and treatments, and the final phase, which includes activities performed in cases of ulcer healing. In the case of ulcer healing in 3 months, the total cost of the patient management was estimated at EUR 1892, of which EUR 302 (16%) was borne by the patient.

After the stenting procedure, the patient was prescribed compression therapy to facilitate the healing process. Therefore, in the CEA model, the follow-up costs of SOC were included also for the stenting strategy (conservative approach).

### Cost-Effectiveness Analysis

QALYs and costs were obtained through the model considering a time horizon of 3 years. The mean QALYs for stenting and SOC strategies were 2.679 and 2.498, respectively. From the hospital perspective, the mean cost per patient for stenting strategy and SOC was EUR 8731 and EUR 6871. The ICUR of stenting compared to SOC was EUR 10,270/QALY. However, from the societal perspective (considering also costs borne by the patients), the mean cost per patient for stenting strategy and SOC was EUR 9204 and EUR 7581, respectively, leading to an ICUR of EUR 8962/QALY.

One-way sensitivity analyses for the ICUR for the hospital perspective are reported in Table 3. The parameters reporting greater variations on the ICUR were the percent of ulcers healed for stenting and SOC, the utility coefficient for a healed and active ulcer, and the cost of the stenting procedure. The same conclusions are valid from the societal perspective (data not shown).

Figure 2 reports the PSA in the form of an acceptability curve, which shows that for a willingness to pay threshold greater than EUR 10,000, stenting may be considered a cost-effective option compared to SOC from the hospital perspective. 

Figure 3 reports the outcome metrics and cost data on a single radar chart diagram for stenting and SOC. The cost borne by the hospital was lower than SOC when compared to stenting, while the cost borne by the patient was lower for the stenting strategy. Clinical outcomes such as ulcer recurrences or healing were better for stenting to improve the overall patient’s quality of life. The greater area of the surface, the better strategy outcomes, as shown for stenting in this specific case.

## 4. Discussion

Deep vein thrombosis, with its thromboembolic complications (e.g., pulmonary embolism), is a serious and potentially fatal disease. This often complicates the clinical course of patients with other diseases, whether already hospitalized or not, and affects patients in apparent good health conditions. The most important clinical objectives of a timely and correct diagnosis and treatment are the reduction of morbidity and mortality associated with its acute manifestations and the reduction of the incidence of relapses or distant sequelae represented by post-thrombotic syndrome, which is often highly disabling and has high healthcare and social costs [30]. Venous leg ulcers related to venous disease account for annual direct medical costs in the range USD 894–USD 10,169 (mean cost of USD 5527) per person per year for patients managed conservatively in Australia, France, Germany, Italy, Spain, UK, and the US [31].

The burden of deep vein thrombosis, and in general of venous diseases, should be evaluated in the current era, in which the healthcare systems worldwide are under great pressure (e.g., due to the lack of healthcare resources, the increase of chronicity incidence, the effect of COVID-19) and the need to reconsider the paradigms related to healthcare management has become increasingly evident. This is especially true for Italy, where the universalistic system represents one of the most precious assets for citizens. In this context, the paradigm of value-based healthcare, which focuses on people’s healthcare needs, provides the direction for addressing the present and future challenges. The adoption of the TDABC methodology, whose interest and use is growing [32], makes the process steps involved in a patient cycle of care more transparent to clinicians, allowing to better estimate the time and the resources absorbed. Moreover, TDABC allows an analysis of the total cost of care cycle and, consequently, extends beyond the traditional reimbursement schemes, which promotes an integrated view of the process steps among different hospital contexts and/or settings. In this perspective, a full cost model strictly linked to the activities implemented (including their physiological variability) may represent an added value in cost-effectiveness analyses. 

The context of the present study is the cost-effectiveness evaluation of medical device technologies, namely endovascular stenting and SOC for managing patients with DVT and leg ulcers, through the use of the TDABC methodology. Medical devices show particular challenges for health technology assessments caused by rapid innovation, outcomes influenced by training, the competence of final users, and dynamic pricing [33]. SOC, which consists in compression therapy, with or without anticoagulation, is considered the standard conservative management for patients with deep venous obstruction and venous ulceration [34]. On the other side, the endovascular approach through stenting has become a broadly accepted treatment strategy in chronic venous obstruction reporting minimal complications, high technical success rates, and short hospital stay [35,36]. In this context, the assessment of the cost-effectiveness of this innovative technology takes on importance in the view to pursue the paradigm of “value-based healthcare”.

For endovascular stenting, we identified 9 macro-phases and 36 activities of the patient care cycle. For each of these steps, we gathered timing data and identified resources activated. These steps enabled us to calculate the cost capacity rates for each capacity resource, with a total average cost per patient equaling EUR 5234. Similarly, we calculated the costs for SOC at EUR 1892, which includes the total cost of the hospital and the patient, for the management of an ulcer that heals in 3 months. These cost values were then used to evaluate the cost-effectiveness of the treatments included in the study, confirming that venous stenting may be a cost-effective strategy compared to SOC for the management of patients with DVT and leg ulcers from the hospital and societal perspectives in Italy. 

On the other hand, the analyses showed that the hospital cost for stenting (EUR 5234) is higher than the DRG reimbursement at national level (EUR 4742). Concerning SOC, the patient management consists of different outpatient activities that are reimbursed according to outpatient tariffs, which vary across Italian regions. For a patient with an ulcer healing in 3 months, a total of 12 visits and medications (including cleaning, possible debriding, disinfection, etc.) are performed. This leads to reimbursement of about EUR 1132 with a mean cost of medications of EUR 72 (we referred to the specific tariffs for Lombardy region) (visit reimbursement EUR 22). Given the hospital cost of EUR 1590 for managing a healing ulcer in 3 months, again the reimbursement may not cover the real costs sustained at the clinical center. Policies that promote the dispensation of therapies and bandages may increase the patients’ compliance leading to better clinical and patient outcomes. 

Another aspect that deserves mentioning is the further advantages that may be explored with the data collected. The TDABC methodology offers a view on the dimension of production capacity, thus overcoming the “black-box” perspective of stenting procedure, and isolates value-added activities for clinicians and nurses’ effort during the hospitalization, thus, gaining valuable knowledge to improve patient management and experience. Moreover, TDABC permits the management and reduction of clinical risks for the patient during each care process step (e.g., waiting time in the operating room before the starting of surgical intervention) and may identify process improvement opportunities (e.g., personnel change, workflow modification, facility/volume variation) allowing possible costs reduction.

The present study has limitations that need to be recognized. First, the analysis did not consider the general costs of the clinical center (e.g., administrative costs), so the total cost of stenting procedure may be underestimated. 

Second, differently from previous studies, the TDABC approach takes into consideration a few aspects related to the organizational impact on the construction of the endovascular surgical unit in a clinical center (e.g., spaces, equipment, dedicated personnel). However, it did not consider substantial initial investments (e.g., creation of a multidisciplinary team, training) and specific procurement choices of the single hospital (such as the use of a “loan for use agreement”) to treat leg ulcers with endovascular stenting procedure [37]. Additionally, this aspect could have underestimated the total cost of the procedure. 

Third, the analysis on stenting considered a rate of technical success assessed through a meta-analysis, anyway this data is influenced by the experience of the operators (learning curve). Additionally, the centers performing a higher volume of procedures may obtain better performance of the device, better health outcomes, and lower procedure costs [38]. 

The TDABC for stenting was performed at Hesperia Hospital, the center in Italy presenting the largest case series. It is a private hospital that performs stenting procedures efficiently, with an important attention in the process management. It is also a specialist hospital with a limited number of hospital beds (i.e., 100). On the other side, public hospitals may underestimate the bridging role of lean tools and logic. Additionally, these centers register more complex cases due to the higher number of hospital beds and the generalist nature of hospitals in which stenting procedure is performed. Moreover, public hospitals manage fewer leg ulcers cases than Hesperia Hospital, and this may reduce the efficacy of clinical competence and experience. The extension of the study to public hospitals could highlight these differences, cover this information gap and provide a validation of the study results in a broader setting. Continuous data collection and monitoring could provide more robust data also for evaluating these aspects. 

Over a 3-year time horizon, the cost borne by the patients for the purchase of treatments or compression systems is relatively high (EUR 473 for stenting strategy and EUR 711 for SOC). The analysis from the societal perspective was limited to costs sustained by the patients (the so-called out-of-pocket costs) and did not take into consideration possible productivity losses due to the disease. For this reason, the costs borne by the patients may be underestimated. 

## 5. Conclusions

From a health economics perspective, the TDABC methodology permits overcoming the difference between “costs” and “public expenditure”. This methodology applied to stenting and SOC for the management of patients with DVT and leg ulcers showed that venous stenting may be a cost-effective option compared with SOC but that reimbursement rates may not completely cover the real costs sustained by hospitals and clinical centers for the management of these patients. Moreover, the cost for the management of a leg ulcer is partially sustained by the patients themselves. The study underlined how TDABC supports the management of complex costing of hospital settings to create value in healthcare. In this context, the implementation of a more efficient policy for covering the real costs may be beneficial for both clinical centers and patients.

## Figures and Tables

**Figure 1 ijerph-20-05817-f001:**
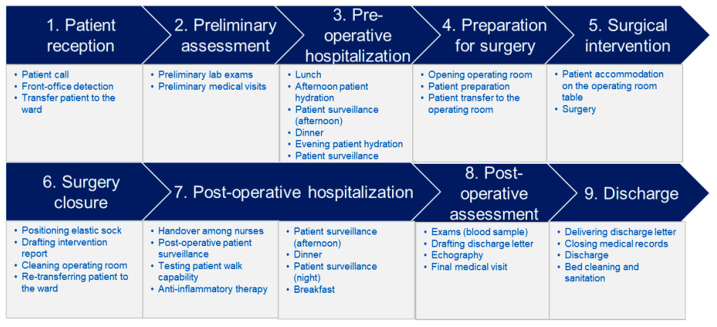
Detail of the activities for the different 9 macro-phases for stenting.

**Figure 2 ijerph-20-05817-f002:**
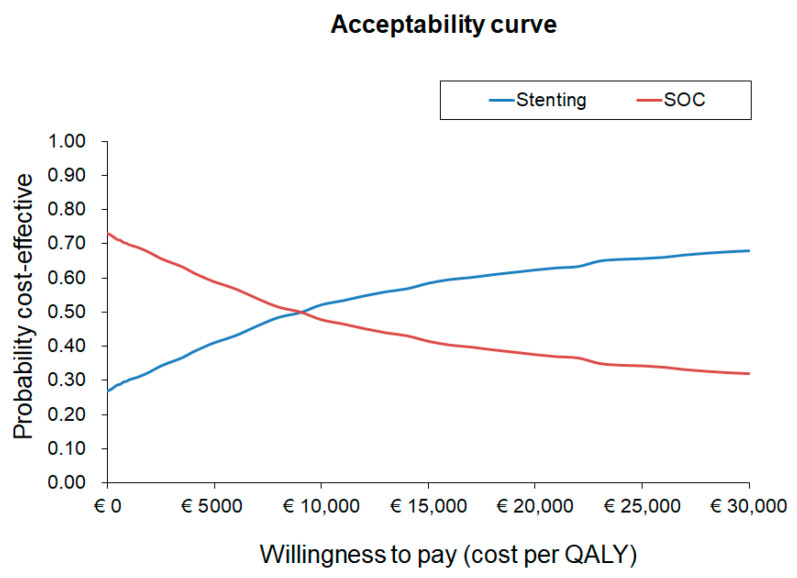
Acceptability curve for the ICUR of stenting vs. SOC (hospital perspective).

**Figure 3 ijerph-20-05817-f003:**
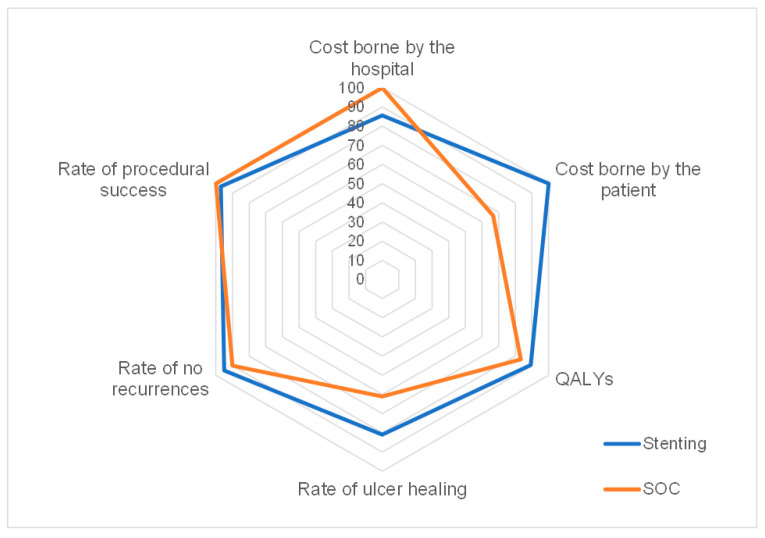
Radar chart plot of outcome and cost metrics for stenting and SOC.

**Table 1 ijerph-20-05817-t001:** Cost components for stenting process.

Macro-Phase	Timing (in Minutes)	Timing (%)	Personnel Costs (EUR)	Good, Services, Tech and Infrastructural Costs (EUR)	Total Costs (EUR)	Total Costs (%)
1 Patient reception	21	1%	13.42	0.67	14.09	0%
2 Preliminary assessment	127	4%	185.82	239.56	425.38	8%
3 Pre-operative hospitalization	1161	39%	106.07	37.16	143.22	3%
4 Preparation for surgery	87	3%	70.07	10.48	80.56	2%
5 Surgical intervention	145	5%	479.67	3519.51	3999.18	79%
6 Surgery closure	80	3%	92.31	10.85	103.16	2%
7 Post-operative hospitalization	1255	42%	144.66	40.16	184.82	4%
8 Post-operative assessment	35	1%	50.87	1.12	51.99	1%
9 Discharge	72	2%	77.27	2.30	79.58	2%
Total	2983	100%	1220.17	3861.82	5081.99	100%

**Table 2 ijerph-20-05817-t002:** Costs for the different phases of SOC.

Phase	Timing (in Minutes)	Personnel Costs (EUR)	Good, Services, Tech and Infrastructural Costs (EUR)	Cost Borne by the Patient (EUR)	Total Costs (EUR)
*First visit*					
Patient reception	2	0.73	0.02	-	0.75
Acquisition of informed consent	3	1.10	0.03	-	1.13
Dermatological visit	65	51.16	58.59	-	109.75
Communication to patient and caregiver	20	32.32	0.11	-	32.43
Definition of treatment program *	15	24.24	67.38	-	91.62
Delivery of treatment program *	15	8.10	0.08	-	8.18
Total	120	117.65	126.21	-	243.86
*Subsequent visits and treatments (each month)*					
Patient reception	2	0.73	0.02	-	0.75
Visit	20	32.32	0.32	-	32.64
Cleansing, curettage and swab	20	21.56	3.11	-	24.67
Wound debridement ****	20	32.32	8.95	-	41.27
Medication **	5	2.70	3.36	-	6.06
Treatment and bandages	107	57.66	241.43	38.72	337.80
Booking next visit	3	1.62	0.02	-	1.64
Total	176.8	148.91	257.21	38.72	444.83
*Final phase (in case of ulcer healing)*					
Last medical visit	20	32.32	0.11	-	32.43
Instruction on braces use	10	5.40	0.05	-	5.45
Closing therapy ***			25.00	185.50	210.50
Control visits	40	64.64	0.43	-	65.07
Total	70	102.36	25.59	185.50	313.45

* The program is defined during the first visit, but its delivery to the patient takes place as soon as the clinical situation is totally clear (e.g., post-consultation among professionals); thus, with a lag of a few days, involving only the nursing component; ** It refers to the wound cleaning; *** refers to the use of antithrombotic drugs and elastic socks used by patient before the last two control visits; **** Frequency: 50% of visits.

**Table 3 ijerph-20-05817-t003:** One-way sensitivity analyses on the ICUR of stenting vs. SOC (hospital perspective, base case ICUR = EUR 10,270/QALY).

Parameter	Lower Limit	Base Case	Upper Limit	Lower ICUR Limit (EUR)	Upper ICUR Limit (EUR)	Variation (EUR)
Stenting % ulcers healed	0.65	0.81	0.97	62,363.85	Dominant	CEA result change
SOC % ulcers healed	0.49	0.61	0.73	1436.29	33,566.77	32,130.48
Utility healed ulcer	0.80	1.00	1.00	36,040.67	10,270.18	25,770.49
Utility active ulcer	0.58	0.73	0.88	7007.05	19,221.46	12,214.41
Cost Stenting	4187.20	5234.00	6280.80	4489.06	16,051.30	11,562.24
Cost FUP—active ulcer (monthly)	324.89	406.11	487.33	13,754.31	6786.05	6968.26
SOC mean healing time (months)	2.40	3.00	3.60	11,016.75	9549.95	1466.80
Utility recurred ulcer	0.51	0.64	0.77	9787.21	10,803.29	1016.08
Cost first visit—active ulcer	195.09	243.86	292.63	10,540.74	9999.62	541.12

## Data Availability

The data that support the findings of this study are available from the corresponding author upon reasonable request.

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
