# Peer review of "Time-Driven Activity-Based Costing for Capturing the Complexity of Healthcare Processes: The Case of Deep Vein Thrombosis and Leg Ulcers"

_ijerph, 2023, doi:10.3390/ijerph20105817_

Round 1

Reviewer 1 Report

  • The paper "Time Driven Activity Based Costing for Capturing the Complexity of Healthcare Processes: the Case of Deep Vein Thrombosis and Leg Ulcers" is well structured and aims to provide information to analyze cost-efficacy of different treatment of Deep Vein Thrombosis and Leg Ulcers.

The analysis is very deep and meaningful. Each step is clearly described.

To make the analysis more solid, it is useful to provide a few more details on the calculation of personnel costs (sources and methods) also because they are linked to different roles. Perhaps some details could be added in supplementary materials also to enable benchmark or to repeat the study in different hospitals or scenarios.

This is also because tables 1 and 2 seem to hide this detail and risk being misunderstood. Alternatively, at least, a note can explain that personnel costs are computed basing the time spent in each phase by each role.

The attention to the process management cited at row 380, is important to contextualize the opportunity of further costs reduction that could be a discussion point.

In the conclusion, maybe could be useful to provide some limitations of this study: it needs some validation from other Institutions? Statistical limits?

I think that the paper is a relevant example of application of the methodology for the definition of costs in specific healthcare process and also a good discussion template.

The revisions I suggest are really minor revisions, and authors might consider them as conceptual suggestions to improve readability and reader understanding.

Author Response

Dear Editor-in-Chief and Reviewers

First of all, we would like to thank the reviewers for their constructive comments and suggestions. Please find below the required changes (marked with R:) in response to the reviewer’s remarks for your consideration. We have carefully revised the manuscript and we hope that the changes made improved the quality of the manuscript and made it suitable for publication.

Best Regards,

Carla Rognoni on behalf of all authors

Comments and Suggestions for Authors

The paper "Time Driven Activity Based Costing for Capturing the Complexity of Healthcare Processes: the Case of Deep Vein Thrombosis and Leg Ulcers" is well structured and aims to provide information to analyze cost-efficacy of different treatment of Deep Vein Thrombosis and Leg Ulcers.

The analysis is very deep and meaningful. Each step is clearly described.

To make the analysis more solid, it is useful to provide a few more details on the calculation of personnel costs (sources and methods) also because they are linked to different roles. Perhaps some details could be added in supplementary materials also to enable benchmark or to repeat the study in different hospitals or scenarios.

This is also because tables 1 and 2 seem to hide this detail and risk being misunderstood. Alternatively, at least, a note can explain that personnel costs are computed basing the time spent in each phase by each role.

R: We thank the Reviewer for pointing out this lack. We included a description on the estimation of personnel costs and a supplementary table (new Supplementary Table 3) with the details:

“For healthcare personnel, costs were computed by multiplying the minutes spent in each phase by the wage per minute for each role (details about the capacity costs for the different roles and timings applied are reported in Supplementary Table S3).”

Also for SOC we reported:

“(see Supplementary Table S3 for details on healthcare personnel costs)”

The attention to the process management cited at row 380, is important to contextualize the opportunity of further costs reduction that could be a discussion point.

R: We reported a description to better contextualize the discussion:

“Another aspect that deserves mentioning is the further advantages that may be explored with the data collected. The TDABC methodology offers a view on the dimension of production capacity, thus overcoming the “black-box” perspective of stenting procedure, and isolates value-added activities for clinicians and nurses’ effort during the hospitalization, thus, gaining valuable knowledge to improve patient management and experience. Moreover, TDABC permits the management and reduction of clinical risks for the patient during each care process step (e.g., waiting time in the operating room before the starting of surgical intervention) and may identify process improvement opportunities (e.g., personnel change, workflow modification, facility/volume variation) allowing possible costs reduction.”

In the conclusion, maybe could be useful to provide some limitations of this study: it needs some validation from other Institutions? Statistical limits?

R: We thank the Reviewer for the suggestion. We reformatted the discussion and explicitly reported a section at the end with the limitations of the study. We included also the aspect related to the validation from other institutions. There are no particular statistical limits since the data were collected through on-site investigation at Hesperia Hospital. There may be limitations in results generalizability but these limitations are already recognized in the paper.

I think that the paper is a relevant example of application of the methodology for the definition of costs in specific healthcare process and also a good discussion template.

The revisions I suggest are really minor revisions, and authors might consider them as conceptual suggestions to improve readability and reader understanding.

R: We thank the reviewer for the positive evaluation and for the useful suggestions. We tried to improve the paper accordingly.

Reviewer 2 Report

Dear authors,

thank you very much for the opportunitiy to review your interesting manuscript.

Please find suggestions for improvement:

roe 23 -  what i DRg reimburssement

row 50 - please add reference

row 62 - please decsribe types of endovenous stenting. Also, it would be beneficial fro reader to mre detail explain what is reimbursed in Italy and what is on patient cost

row 107 - please finish/explain the sentence

rows 277-279 - please explain how did you get this calculation

rows 302-347 - this should be shorttened as you repaet something form introduction 

discussion - please define indications for stenting and for standard of care, I thonk that stenting is not always indicated, so why comparing these 2 procedures?

Thank you for your answers.

Author Response

Dear Editor-in-Chief and Reviewers

First of all, we would like to thank the reviewers for their constructive comments and suggestions. Please find below the required changes (marked with R:) in response to the reviewer’s remarks for your consideration. We have carefully revised the manuscript and we hope that the changes made improved the quality of the manuscript and made it suitable for publication.

Best Regards,

Carla Rognoni on behalf of all authors

Comments and Suggestions for Authors

Dear authors,

thank you very much for the opportunity to review your interesting manuscript.

Please find suggestions for improvement:

row 23 -  what is DRg reimbursement

R: We thank the Reviewer for highlighting this unclear point. We specified the meaning of the acronym: Diagnosis-Related Group

row 50 - please add reference

R: Thanks, we included the reference (n. 5).

row 62 - please describe types of endovenous stenting. Also, it would be beneficial for reader to more detail explain what is reimbursed in Italy and what is on patient cost

R: We thank the Reviewer for pointing out this aspect. We included a description of endovenous stenting and gave a brief overview about the different types of stents:

“An endovenous stent may be defined as a synthetic tubular structure implanted in native or graft vasculature to provide mechanical radial support and enhance vessel patency. Actually, more than ten dedicated venous stents are available on the European market. The main differences are related to the design, the material, the deployment system and the different sizes [9-15]. The design can be closed or open cells with different shape and size of the cell area. The material used in new generation venous stents is nitinol or even elgiloy (material composed of cobalt, chromium, nitinol, iron, etc.). Another technical feature is the different modality of deployment, due to guarantee the precision and stability of the stent during the deployment phase in the vein: it can be performed in a pull-back technique or using a rotating thumbwheel; some systems are even re-constrainable, allowing to change and adjust the landing zone for the stent. A wide range of sizes is actually available (10-24 mm diameter, 40-160 mm length).”

Regarding the reimbursement in Italy, we added two paragraphs reporting information on stenting procedure and, for completeness, for compression devices:

 “In Italy endovascular stenting for DVT and leg ulcers is reimbursed by the NHS through DRG (Diagnosis-Related Group) 479 with a tariff at national level of 4,742€ [16].”

“Compression combined with anticoagulant therapy is the standard of care (SOC) for venous ulcer management……. In Italy these products are not reimbursed by the NHS so they are bought by the patients themselves [8].”

row 107 - please finish/explain the sentence

R: Thanks, the sentence has been corrected in:

“For the cost-effectiveness analysis we used the model structure already presented by Rognoni and colleagues [23]”

rows 277-279 - please explain how did you get this calculation

R: We thank the Reviewer for pointing out this unclear point. As we described in the Methods section (paragraph “Cost-effectiveness model”), we used the structure of a model already used in another publication and updated it with clinical data (rates of ulcers healed and recurred obtained from updated meta-analysis) and costs obtained by the application of time driven activity-based costing methodology (instead of data from the literature). We tried to better clarify the description.

Moreover, we added in the Methods the following text “The model allowed to estimate mean costs and QALYs for the management of a patient with the two options, stenting procedure or SOC, over a time horizon of 36 months.”, to clarify that the ICUR has been calculated from costs and QALYs estimated through the model:

“Cost-effectiveness analysis was expressed through the incremental cost-utility ratio (ICUR), calculated as the difference in the mean expected costs divided by the difference in the mean expected QALYs between stenting procedure and SOC, as obtained by the model:

 ICUR = (Cost Stenting – Cost SOC)/(QALYs Stenting – QALYs SOC)”

In the Results section (Cost-effectiveness analysis) we recalled this aspect:

“QALYs and costs were obtained through the model considering a time horizon of 3 years.”

rows 302-347 - this should be shorttened as you repaet something form introduction 

R: We thank the Reviewer for this observation. We shortened the introductory part of the Discussion deleting the concepts already presented in the Introduction.

discussion - please define indications for stenting and for standard of care, I think that stenting is not always indicated, so why comparing these 2 procedures?

R: We thank the Reviewer for highlighting this important point. In the Discussion we included a rationale for the comparison:

“SOC, which consists in compression therapy, with or without anticoagulation, is considered the standard conservative management for patients with deep venous obstruction and venous ulceration [34]. On the other side, the endovascular approach through stenting has become a broadly accepted treatment strategy in chronic venous obstruction reporting minimal complications, high technical success rates and short hospital stay [35,36]. In this context, the assessment of the cost-effectiveness of this innovative technology takes on importance in the view to pursue the paradigm of “value-based healthcare”.”

Thank you for your answers.

Round 2

Reviewer 2 Report

Dear authors.

Thank you very much for your effort in correcting the manuscript. I find it more readable in usable for readers, which is actually the point of publishing. Good luck with future publishing!